# Clinical-Demographic Profile, Prognostic Factors and Outcomes in Classic Follicular Lymphoma Stratified by Staging and Tumor Burden: Real-World Evidence from a Large Latin American Cohort

**DOI:** 10.3390/cancers16233914

**Published:** 2024-11-22

**Authors:** Daniel Silva Nogueira, Luís Alberto de Pádua Covas Lage, Cadiele Oliana Reichert, Hebert Fabrício Culler, Fábio Alessandro de Freitas, João Antônio Tavares Mendes, Ana Carolina Maia Gouveia, Renata de Oliveira Costa, Cristiane Rúbia Ferreira, Jéssica Ruivo Maximino, Sérgio Paulo Bydlowski, Carlos Alejandro Murga Zamalloa, Vanderson Rocha, Débora Levy, Juliana Pereira

**Affiliations:** 1Department of Hematology, Hemotherapy and Cell Therapy, Faculty of Medicine, University of Sao Paulo (FM-USP), Sao Paulo 05508-090, Brazil; nogueira.hemato@gmail.com (D.S.N.); cadiele.reichert@hc.fm.usp.br (C.O.R.); hebert.culler@hc.fm.usp.br (H.F.C.); fabio.alessandro@hc.fm.usp.br (F.A.d.F.); spbydlow@usp.br (S.P.B.); vanderson.rocha@hc.fm.usp.br (V.R.); juliana.pereira@hc.fm.usp.br (J.P.); 2Laboratory of Medical Investigation in Pathogenesis and Directed Therapy in Onco-Immuno-Hematology (LIM-31), University of Sao Paulo (FM-USP), Sao Paulo 05508-090, Brazil; joaoantonio15@hotmail.com (J.A.T.M.); ana.gouveia@fm.usp.br (A.C.M.G.); 3Department of Hematology, Faculty of Medicine, Centro Universitário Lusíada (Unilus), Santos 11050-071, Brazil; renatadeoliveiracosta@uol.com.br; 4Department of Pathology, University Hospital (HU), University of Sao Paulo (FM-USP), Sao Paulo 05508-090, Brazil; rubia082@gmail.com; 5Laboratory of Translational Neurology (LIM-45), University of Sao Paulo (FM-USP), Sao Paulo 05508-090, Brazil; jessica.maximino@fm.usp.br; 6Laboratory of Immunology and Histocompatibility (LIM-19), University of Sao Paulo (FM-USP), Sao Paulo 05508-090, Brazil; d.levy@hc.fm.usp.br; 7Department of Pathology, University of Illinois at Chicago (UIC), Chicago, IL 60607, USA; catto@uic.edu; 8Fundação Pró-Sangue, Blood Bank of Sao Paulo, Sao Paulo 05468-901, Brazil; 9Department of Hematology and Hemotherapy, Churchill Hospital, Oxford University, Oxford OX1 2JD, UK; 10Department of Hematology and Oncology, Hospital Alemão Oswaldo Cruz (HAOC), Sao Paulo 01323-020, Brazil

**Keywords:** follicular lymphoma (FL), staging, tumor burden, prognostic factors, progression of disease within 24 months from initial therapy (POD-24), histological transformation (HT) to high-grade B-cell lymphoma (HGBCL), clinical outcomes, treatment modalities

## Abstract

Follicular lymphoma (FL) is an indolent heterogeneous B-cell malignancy with variable clinical presentation and outcomes. Despite previous prognostic clinical scores’ validation, unmet medical needs remain in understanding the reason why selected groups of FL patients experience a less or more aggressive clinical course with higher risk of relapse, death or histological transformation (HT). Prognostic refinement by stratification of symptomatic disease and clinical evolution according to staging and tumor burden status are crucial gaps that requires further investigation. Therefore, this study will present advances in clinical subclassification and in the impact of staging and tumor burden on treatment efficacy and survival. Additionally, it will provide new perspectives on characterization of risk factors for HT and early-progression (POD-24). Insights into disease progression patterns may help clinicians to establish personalized treatment strategies for FL patients.

## 1. Introduction

Follicular lymphoma (FL) is a low-grade malignancy characterized by germinal-center B-cell proliferation with nodular arrangement, presence of the recurrent chromosomal abnormality t(14;18) (q32;q21) and overexpression of the anti-apoptotic protein *bcl-2* [1]. This subtype of indolent non-Hodgkin lymphoma (NHL) presents marked biological heterogeneity and variable clinicopathological features [2]. It represents approximately 25–35% of all NHLs and most frequently affects older patients, with a median incidence of 65 years at diagnosis [3].

Currently, the 5th edition of the World Health Organization classification for Hematopoietic and Lymphoid Tissue Tumors (WHO-HAEM 5th) recognizes four main subtypes of FL, including the classic follicular lymphoma (cFL), in situ follicular B-cell neoplasm, duodenal-type FL, and pediatric-type FL [1]. Apart from cFL, WHO-HAEM 5th. also recognizes the entities “FL with unusual cytological features” (uFL), “FL with a predominantly diffuse growth pattern” (dFL), and “follicular large B-cell lymphoma” (FLBCL) [1]. Usually, the cFL’s malignant cells constitute a mixture of centrocytes and centro-blasts in variable proportions, expressing pan B-cell antigens, such as CD19, CD20, CD22, CD79b, and PAX-5, germinal-center markers, such as CD10 and BCL-6, and the anti-apoptotic protein BCL-2, but not the antigens CD5 and CD23. However, recently FL entities have been described with absence of CD10 or BCL-2 expression, particularly the subtypes pediatric and primary cutaneous [4]. Despite the prototypical phenotypic and molecular-genetic findings, other types of FL have been recognized, including lymphoma *BCL-2*-rearranged-negative (*BCL2*-R-negative), CD23-positive follicular center lymphoma, pediatric-type FL and primary cutaneous centro-follicular lymphoma [4,5,6].

Previously, the 3rd and 4th WHO editions [7,8] graduated FL based on its histopathology according to the number of centro-blasts (CB) per high-power field (HPF). Grade 1 was defined by <5 CB/HPF, grade 2 by 5 to 15 CB/HPF, and grade 3 by >15 CB/HPF, and the latter was subdivided into grade 3A and 3B. To characterize grade 3B, the neoplastic tissue must be composed of solid sheets of CB, with absence of residual centrocytes, a morphological figure virtually indistinguishable from diffuse large B-cell lymphoma (DLBCL). Nowadays, it is understood that FL grade 3B must be treated as DLBCL [9,10,11]. However, the treatment for grades 1, 2 and 3A is not distinct [12,13]. Based on this premise, and considering the lack of reproducibility to differentiate grades 1, 2 and 3A, in the 5th edition of the WHO classification, FL graduation became optional [1]. Therefore, FL grades 1–3A were gathered as classic FL, while FL grade 3B was renamed as follicular large B-cell lymphoma (FLBCL) [5]. Not infrequently, FL grade 3B can be not CD10 and/or BCL-6 negative [5]. Additionally, the WHO 5th edition also recommended the name “FL with unusual cytological features” (uFL) for morphological variants, such as those characterized by large centrocytes or blastoid morphology, and “FL with a predominantly diffuse growth pattern” (dFL) for those with a predominantly diffuse growth pattern rather than follicular/nodular [1].

The biology of cFL is centered on the presence of the recurrent genetic abnormality t(14;18)(q32;q21). *IgVH:BCL-2* translocation leads to *BCL-2* gene rearrangement in 85–90% of cases [14]. The *bcl-2* protein has an antiapoptotic function and consequently prolongs the lifespan of mutated B-cells in the germinal center, making them more susceptible to the acquisition of secondary clonogenic alterations, associated with disease progression and histological transformation (HT) to high-grade B-cell lymphoma (HGBCL), a hallmark of FL’s natural history. However, 10–15% of FL cases, especially grade 3B [15], remain BCL-2 negative, since they lack *BCL-2*-R [16]. This can also be justified due to the fact that certain *BCL-2* gene mutations hinder recognition by the monoclonal antibodies conventionally used for routine staining by immunohistochemistry [17,18,19]. Moreover, mutations in epigenetic regulators and chromatin-remodeling genes, such as *KMT2D*, *MLL2*, *CREBBP*, and *EP300*, are common in FL [20]. These genes also represent early drivers of FL. Mutations of *EZH2* and epigenetic dysregulation occur in 25% of cases and have been used as a therapeutic target in clinical trials [21]. Less frequently, epigenetic modifiers mutations may occur in *ARID1A*, *MEF2B* and *KMT2C* genes. Genetic abnormalities can also be found in genes involved in cell signaling, such as *STAT6*, *CARD11* and *FOXO1*, and alterations in *TP53*, *CDKN2A* and *MYC* are usually associated with high-grade features and increased risk of transformation to HGBCL [22].

FL is still considered an incurable malignancy, presenting progression-free survival (PFS) of 40% to 80% at 10 years [12,23]. The main risk stratification is associated with clinical staging (CS) according to Ann Arbor/Cotswold classification, and the tumor burden (TB) at diagnosis [24,25,26]. These characteristics also help to guide therapy [13,24,26,27,28,29]. Generally, patients with FL stage I and II with non-bulky disease (early-stage/ES) may be treated with isolated involved-field radiotherapy (IF-RT) at low-doses (20–24 Gy), anti-CD20 immunotherapy, or via a strategy of watching and waiting (WW) [13,30]. Patients with stage III or IV (advanced disease) with low tumor burden or asymptomatic (AS-LTB) can be kept preferably in WW or treated with isolated immunotherapy. On the other hand, FL patients with stage III or IV presenting symptomatic disease and/or higher tumor burden (AS-HTB) should be treated with immunochemotherapy (ICT), using regimens such as R-CHOP (rituximab, cyclophosphamide, doxorubicin, vincristine and prednisone), R-CVP (rituximab, cyclophosphamide, vincristine and prednisone), R-FC(M) (rituximab, fludarabine, cyclophosphamide with or without mitoxantrone) or BR (bendamustine plus rituximab) [27,31,32].

Clinical prognostic scores, such as the indexes FLIPI (*Follicular Lymphoma International Prognostic Index*) and FLIPI-2 (*Follicular Lymphoma International Prognostic Index-2*), have been proposed to stratify prognosis in FL, but they are not used to guide therapy. Progression of disease within 24 months after first treatment (POD-24) is associated with lower overall survival (OS), but can be identified only after the treatment [33,34]. POD-24 is a crucial prognostic factor for AS-HTB patients, currently becoming a strong marker of shortened survival, and it indicates the need for intensified therapy, preferably including consolidation based on autologous stem cell transplantation (ASCT) [35]. Later, other authors proposed scores based on the gene profile [36] of tumor cells, such as the 7-gene-clinicogenetic model (m7-FLIPI), which improved the risk stratification of FL in patients receiving ICT [20]. However, their prognostic score failed to demonstrate impact in patients conducted with chemo-free regimens, such as R2 (rituximab plus lenalidomide) [37]. Another study, also using the gene-expression profile, proposed the 23-gene model capable of predicting progression of disease [38], but it was not able to stratify different groups of treatment, and other studies are necessary to clarify the prognostic role of the gene signature in FL [39,40]. Therefore, despite the broad biological advances obtained in the last decade, the prognostic stratification and guide of treatment in FL remain based on clinical aspects, including staging, presence of symptoms and tumor burden.

Based on this premise, the present study aims to describe clinical and laboratory characteristics, assess outcomes, determine predictors of survival and HT to HGBCL, and compare responses between different primary therapeutic strategies applied in a large cohort of Brazilian patients with FL categorized according to clinical staging and tumor burden.

## 2. Materials and Methods

### 2.1. Study Design and Ethical Issues

This is an observational, retrospective, and single-center study carried out at the Instituto do Câncer do Estado de São Paulo (ICESP)/Hospital das Clínicas, Faculty of Medicine, University of São Paulo (HC-FMUSP), Brazil. The study was conducted in accordance with the principles of the Declaration of Helsinki. Ethical approval was obtained from the appropriate institutional review board in 2023 (CAAE number: 32830220.8.0000.0068). Application of the Free and Informed Consent Form (FICF) was waived by the institutional review board due to the retrospective nature of the study. After institutional Ethics Committee approval, baseline clinical-demographic, laboratory, pathological, imaging and therapeutic data of patients diagnosed with FL grades 1, 2 and 3A between January 2000 to January 2022 registered in our Lymphoma Database were retrospectively captured from medical records and inserted into *Research Electronic Data Capture (RedCap)* platform.

### 2.2. Histopathological Diagnosis

In parallel, the available biopsy samples obtained at diagnosis were recovered and centrally reviewed by an expert in Hematopathology and were classified according to the 2016 WHO classification of Hematopoietic and Lymphoid Tissue Neoplasms (WHO-2016) updated in 2017 [6,41].

All cases with the initial diagnosis of FL were retrieved from the Archives of the Division of Anatomic Pathology, Faculty of Medicine, University of Sao Paulo (DAP/HC-FMUSP) and from the Institute of Cancer (ICESP) from January 2000 to January 2022. The histopathological review was conducted on formalin-fixed and paraffin-embedded (FFPE) material, from lymph nodes and/or any other extra nodal tissues, stained by Hematoxylin-Eosin (H and E) followed by immunohistochemistry (IHC) study.

The morphological patterns evaluated were follicular and/or diffuse cytoarchitecture, histological graduation 1/2 (less than 15 CB per HPF—40x objective, optic microscopy), 3A (more than 15 CB per HPF with presence of residual centrocytes) and 3B (more than 15 CB per HPF with no residual centrocytes). The presence of large centrocytes was recorded to prevent misinterpretation with grades 3A and 3B. The immunohistochemical expression of CD20 (Dako, L26, 1/1000), CD5 (Invitrogen, 53-7.3, 1/200), CD10 (Novocastra, S6C6, 1/2000), BCL-6 (Abcam, EPR11410-43, 1/500), BCL-2 (Sigma-Aldrich, B3170, 1/500) and MUM-1/IRF-4 (Abcam, EPR5653, 1/500) markers on the neoplastic cells were evaluated, as well as CD23 (Sigma-Aldrich, 1B12, 1/300) in the network of follicular dendritic cells (FDC) and Ki-67 (Dako, J55, 1/1600) index were also annotated.

### 2.3. Patients, Eligibility Criteria, and Staging Procedures

Considering all patients diagnosed with NHL registered in the ICESP/HC-FMUSP Non-Hodgkin’s Lymphoma Database, 228 patients diagnosed with FL were initially selected. After applying the eligibility criteria, 214/228 (93.6%) were included in this study, as summarized in Figure 1. Eligibility criteria included age ≥ 18 years, cases diagnosed and treated at ICESP/HC-FMUSP from January 2000 to January 2022, and biopsy-proven diagnosis of classic FL according to the criteria proposed by the WHO classification for Hematopoietic and Lymphoid Tissue Neoplasms published in 2016 [41]. Patients diagnosed with in situ FL, duodenal-type FL, pediatric FL, primary cutaneous centro-follicular lymphoma, FL with HT to HGBCL at diagnosis (“synchronous transformation”) and those cases with clinical data considered insufficient were excluded from this analysis. After applying the eligibility criteria, 14 (6.4%) cases were excluded, including 5 cases with synchronous HT to HGBCL, 3 cases diagnosed as duodenal-type FL, 3 cases with cutaneous primary centro-follicular lymphoma, and 3 cases with insufficient clinical-laboratory information (Figure 1).

The demographic, clinical and laboratory features evaluated were gender, age, performance status on the *Eastern Cooperative Oncology Group* scale (ECOG) [42], bulky disease considered as tumor mass ≥ 7 cm in its largest diameter, presence of comorbidities, B-symptoms, number and size of nodal and extra nodal areas involved by lymphoma and BM involvement status. Similarly, presence of malignant cells infiltrating pleural or abdominal cavity confirmed by oncotic cytology, flow cytometry and/or biopsy, hemoglobin level, white blood cell (WBC) count, platelets number, neutrophil/lymphocyte ratio (N/Ly), lymphocyte/monocyte ratio (Ly/M), lactate dehydrogenase (LDH) level, patient LDH/LDH upper limit ratio, β-2 micro-globulin (β2M), leukemic phase characterized by lymphocyte count > 5 × 10^9^ cells/L and/or presence of the malignant cells in the peripheral blood confirmed by flow cytometry were recorded. Likewise, presence of immune thrombocytopenia (ITP), autoimmune hemolytic anemia (AIHA), B and C hepatitis and HIV serology were also assessed.

At diagnosis, all patients were submitted to computed tomography (CT) scan of the neck, thorax, abdomen and pelvis, or preferentially CT with positron emission with 18-fluorodeoxyglucose (18FDG-PETCT), and unilateral BM biopsy. When indicated, especially in patients who are candidate for regimens containing anthracycline, transthoracic echocardiogram was also performed. The *Follicular Lymphoma International Prognostic Score* (FLIPI) [43] and FLIPI-2 [44] were also assessed at diagnosis for the whole cohort. Lymphoma staging was performed according to Ann Arbor/Cotswold stage system [45]. After performing the staging procedures, FL patients were categorized in early-stage (ES) (I/II with non-bulky disease) and advanced stage (AS) III and IV. Patients with stage III and IV were subdivided into asymptomatic/low tumor burden (AS-LTB) and symptomatic/high tumor burden (AS-HTB) according to the *Groupe d’Etudes des Lymphomes Folliculaires* (GELF) criteria [29,46,47].

### 2.4. Up-Front Therapy, Response Assessment, and Follow-Up

The treatment strategy was based in our institutional protocols at the time of diagnosis. Patients with non-bulky ES FL were usually submitted to IF-RT with 20–24 Gy or kept in WW strategy. Those with AS-LTB were kept in WW strategy or, when available, included in clinical trials using rituximab monotherapy or biosimilar anti-CD20 monoclonal antibodies. Patients with AS-HTB according to GELF criteria [29] were treated with ICT protocols, including R-CHOP-21, R-CVP-21 or ICT based on purine analogs (ICT-P), including the regimens FCR(M) and FR, as summarized in Table 1. Patients treated before the availability of rituximab at our institution received only chemotherapy, such as CHOP or CVP, but comprised a minimum number of cases included in this study. Other regimens applied with palliative intent included monotherapy with chlorambucil or oral fludarabine. Adverse events related to treatment were graduated using the *Common Terminology Criteria for Adverse Events* (CTCAE version 5.0, 2017) [48].

A pre-phase with CVP (cyclophosphamide 300 mg/sqm I.V. on D1, vincristine 1 mg fixed dose I.V. on D1, and prednisone 40 mg/sqm P.O. on days 1 to 7) was used in frail patients, with ECOG > 2 at diagnosis, Karnofsky Performance Scale (KPS) < 70 points, involvement of the gastrointestinal tract to prevent the risk of perforation or with bulky disease, and risk of tumor lysis syndrome. This cytoreductive regimen was used for just 1 cycle given 7–14 days before the full dose of ICT. Outside the clinical research setting, no patient underwent maintenance immunotherapy due to the unavailability of this therapy in the Brazilian public health system (Sistema Único de Saúde/SUS). Prophylaxis for infection included use of granulocyte growth factors (G-CSFs) for individuals with absolute neutropenia < 1.0 × 10^9^/L, as well as prophylaxis for *Pneumocystis jirovecii* infection with trimethoprim-sulfamethoxazole 400/80 mg P.O. per day and secondary prophylaxis for herpes zoster with acyclovir 400 mg P.O. twice per day. Antibacterial prophylaxis with quinolones was not routinely adopted.

Response was evaluated in patients submitted to treatment according to the Lugano Criterion using CT scan or 18FDG-PETCT after cycle four and at the end of treatment [49,50,51]. After treatment, patients in complete response (CR) or partial response (PR) were followed through clinical examination and laboratory tests every four months in the first year and then every six months. Image tests were indicated in case of signals or symptoms suggestive of disease progression or relapse, or at discretion of the physician. Patients in WW strategy were evaluated with image and laboratory tests every six months in the first year of diagnosis and afterwards in case of stable disease (SD) according to the symptoms or at discretion of the physician. At progression/relapse, a new biopsy was indicated for all cases to confirm it and to verify if there was HT to HGBCL. HT was defined by the presence of FL grade 3B or DLBCL characterized by diffuse sheets of large cells. Despite the fact that the Ki-67 proliferation index is important and has prognostic significance, it was not used as a feature for transformation to HGBCL, according to WHO recommendations [1]. The study design is depicted at Figure 1.

### 2.5. Statistical Analysis

Data were shown in accordance with the variables evaluated. Categorical variables were presented as absolute (N) and relative (%) values. Numerical variables were presented as measures of central tendency (median), dispersion (min-max range; IQR 25–75%), and position. The median follow-up time and the cumulative rate of HT to HGBCL were calculated using the reverse Kaplan–Meier (KM) method and the conventional KM method, respectively. Analysis of overall survival (OS) and progression-free survival (PFS) was performed using the KM method. PFS was calculated from the date of initiation of first-line therapy until evidence of disease progression or death by any cause and OS from the date of diagnosis until death by any cause. Data were censored at the last follow-up. Progression within 24 months (POD-24), defined as disease progression within 24 months after first exposure to treatment [33], was assessed only for patients with AS-HTB.

Analysis to determine predictors for outcomes, including those related with OS, PFS, POD-24 and HT, was performed using Cox’s semiparametric univariate method. Multivariate analysis using the multi-step Cox’s regression model was conducted to determine independent prognostic variables. All variables with a *p*-value ≤ 0.10 identified in univariate analysis were included in the final model for multivariate analysis. The results were presented in hazard ratio (HR), 95% confidence interval (95% CI) and forest plots. All analysis were performed using the *R-statistical software version 0.4.9 for Survminer R package* [52] and a *p*-value ≤ 0.05 was assigned as statistically significant.

## 3. Results

### 3.1. Clinical-Demographic, Laboratorial and Histopathological Features

The clinical-demographic, laboratorial and pathological baseline findings of the 214 patients with cFL are summarized in Table 2. The median age was 60 years (IQR 25–75%: 51–69 years), with 55.6% (119/214) of female. Approximately 12% (25/214) had ES disease (I/II), 13.5% (29/214) were categorized as AS-LTB (III/IV) and 74.7% (160/214) presented AS-HTB (III/IV) at diagnosis. Additionally, 15.4% (33/214) of patients had ECOG ≥ 2, 49.5% (106/214) of cases presented B-symptoms, 44.3% (95/214) had bulky disease ≥ 7 cm and 24.3% (52/214) presented organ damage or extrinsic compression by tumor mass.

Extra nodal involvement was found in 66.3% (142/214) of cases; 49.5% (106/214) showed BM infiltration by lymphoma, 21.5% (46/214) had serous effusions and 11.7% (25/214) presented leukemic phase. Moreover, 42.5% (91/214) had ≥2 extra nodal areas involved by FL, 68.7% (147/214) had ≥4 nodal sites involved by lymphoma and 29.9% (64/214) showed more than 3 nodal areas superior to 3 cm in the largest diameter.

Low-risk FLIPI score was seen in 22.4% (48/214) of cases, intermediate-risk in 28.1% (60/214) and high-risk in 49.5% (106/214). Similarly, FLIPI-2 score of low-risk was found in 15% (32/214) of cases, 40.1% (86/214) had intermediate-risk and 44.9% (96/214) presented high-risk score. Baseline clinical and laboratory variables distribution at diagnosis discriminated into the 3 subgroups according to tumor burden and staging (ES, AS-LTB, AS-HTB) are described in Table 3.

Among the 214 selected cases of FL grades 1–3A, 152 cases (71%) had recoverable histopathological material (HE slides, immunohistochemistry, and/or paraffin blocks). The pathological diagnosis of FL was made through lymph node biopsy in 81.2% (173/214) of cases, BM biopsy in 6.6% of cases (14/214), and in 12.2% (26/214) the diagnosis was made by other extra nodal site biopsy. Approximately 75% of cases (154/213) were originally categorized as histological grade 1/2, and 24.4% (52/213) as histological grade 3A. One case (0.04%) did not have a histopathological grade assigned, originating from a BM sample. Original histopathological data (pre-centralized review) indicated a median Ki-67 expression of 30% (IQR: 15–40%). Originally, positivity for CD10 antigen expression was observed in 91.5% (196/214) of cases, CD20 in 100% of cases, BCL-2 in 94% (189/201), and BCL-6 in 92.1% (140/152) of cases. Out of those 214 cases, 152 (71%) with available histopathological samples were centrally reviewed by an expert in Hematopathology according to the 2016 WHO classification of Hematopoietic and Lymphoid Tissue Neoplasms. After performing the centralized pathological review and grouping the reviewed cases and the original report of those cases with non-recoverable material, most of the cases (89.2%) presented CD10 expression (191/214), 91% (183/201) expressed BCL-2 and 92.1% (140/152) had BCL-6 expression. Histologic grade 1/2 was seen in 74.7% (154/206) of cases and 3A in 25.3% (52/206) at diagnosis.

Ninety-three percent (200/214) of the cases had Ki-67 assessed by immunohistochemistry. Of these, 156/200 (78%) had low Ki-67 (<30%) and 44/200 (22%) were categorized as high Ki-67 (≥30%). In a comparative analysis involving the two groups, patients with Ki-67 ≥30% had worse performance status, represented by the frequency of ECOG ≥ 2 (25% versus 13%, *p* = 0.049), higher incidence of relapse/progression (44% versus 27%, *p* = 0.043) and higher mortality (45% versus 29%, *p* = 0.047), when compared to cases with Ki-67 <30%, respectively. The distribution by gender (*p* = 0.248) and median age (*p* = 0.226) did not differ between both groups, as well as the frequencies of serositis (*p* = 0.653), clinical stage (*p* = 0.120), B-symptoms (*p* = 0.325), BM infiltration (*p* = 0.182), bulky disease (*p* = 0.307), FLIPI categories (*p* = 0.197), HT to HGBCL (*p* = 0.254) and ORR (*p* = 0.555). Additionally, the estimated 5-year OS was 78.4% (95% CI: 71.5–85.3%) for cases with Ki-67 <30% and 60.5% (95% CI: 45.1–75.9%) for patients with Ki-67 ≥ 30%, *p* = 0.015. Similarly, the estimated 5-year PFS were 68.6% (95% CI: 52.7–84.5%) and 56.2% (95% CI: 47.6–64.8%) for cases presenting Ki-67 <30% and Ki-67 ≥30%, respectively, *p* = 0.153.

### 3.2. Clinical Outcomes in the Overall Cohort

With a median follow-up of 8.15 years (95% CI: 7.30–8.98 years), the median OS for the whole cohort (N = 214) was 14.60 years (95% CI: 13.11–16.25 years). The estimated 5-year and 10-year OS were 75.4% (95% CI: 69.6–81.7%) and 59.3% (95% CI: 61.6–68.1%), respectively. The median PFS was 7.20 years (95% CI: 5.01 not reached) for the whole cohort (N = 214). The estimated 5-year and 10-year PFS were 57.2% (95% CI: 50.3–65.2%) and 44.7% (95% CI: 36.5–54.8%), respectively. OS and PFS curves are displayed in Figure 2.

The overall mortality rate (OMR) during the entire follow-up was 33.1% (71/214) (95% CI: 27.1–39.6%), and 1.9% of deaths (4/214) were early-deaths, occurring within the first 100 days of diagnosis. Among the causes of death, 29.5% (21/71) were associated with progressive disease; 25.3% (18/71) were caused by infection; 4.2% (3/71) by cardiovascular disease; 12.6% (9/71) were related to a second neoplasm; and in 28.1% (20/71) the death cause was unknown, usually due to deaths occurring outside our institution. Mortality rate was higher in the AS-HTB group (38.7%; 95% CI: 31.4–46.4%) compared to ES patients (16.0%; 95% CI: 5.6–33.7%) and AS-LTB (17.2%; 95% CI: 6.9–33.7%), *p* = 0.012.

### 3.3. Up-Front Therapeutic Modalities, Responses and Clinical Outcomes Stratified by Staging and Tumor Burden

Concerning clinical outcomes for the whole cohort (N = 214), cFL patients categorized in subgroups according to clinical staging and tumor burden presented statistically significant differences in OS (*p* = 0.006), but not in PFS (*p* = 0.26), as summarized in Figure 3.

#### 3.3.1. Early-Stage (ES) Disease

In the ES subgroup (CS I/II with non-bulky disease), 48% (12/25) of patients were treated at first-line with IF-RT at a dose of 20–24 Gy (intention-to-treat), while 52% (13/25) were conducted via a WW strategy. During the entire follow-up time, 2 patients (8%) were treated in a clinical trial setting, by medical decision. The overall response rate (ORR) and complete response (CR) for ES FL patients treated with IF-RT were 100% (12/12) and 83.3% (10/12), respectively. With a median follow up of 7.9 years (95% CI: 5.1–10.7), progression of disease (PD) occurred in 32% (95% CI: 16.4–51.4%) of ES cases (8/25). Considering all ES FL cases, the median PFS was 9.37 years (95% CI: 6.3 not reached) and the median OS was not reached. The estimated 5-year PFS and OS for ES disease were 77.3% (95% CI: 61.5–97.1%) and 91.4% (95% CI: 80.7–100%), respectively.

#### 3.3.2. Advanced Stage-Low Tumor Burden (AS-LTB) Disease

For FL patients categorized as AS-LTB (CS III/IV, asymptomatic and with no GELF criteria), 82.7% (24/29) were managed via a WW strategy at diagnosis, except for three cases who were included in clinical trials and two cases treated by own preference (“patient desire”). During the follow-up, another six patients were enrolled in clinical trials. With a median follow-up of 10.3 years (95% CI: 9.1–11.3), the PD rate was 44.8% (95% CI: 27.9–62.7%). Additionally, the median PFS was 10.57 years (95% CI: 2.57 not reached) and the median OS was not reached. The estimated 5-year PFS and OS for AS-LTB FL cases were 59.5% (95% CI: 43.6–81.2%) and 88.5% (95% CI: 77.0–100%), respectively. Among 18 cases treated after PD to symptomatic disease/high-tumor burden, 6/18 (33.3%) were submitted to R-CHOP regimen, 6/18 (33.3%) received R-CVP, 3/18 (16.6%) experienced ICT based on purine analogs (ICT-P) and 3/18 (16.6%) were treated with palliative strategies.

#### 3.3.3. Advanced Stage-High Tumor Burden (AS-HTB) Disease

Among 160 FL patients classified as AS-HTB (CS III/IV, symptomatic or presenting any GELF criteria), 97.5% (156/160) were treated with 6 to 8 cycles of systemic regimens. Three patients (1.9%) had early-death due to complications before starting therapy and one (0.6%) was treated out of our institution. Induction therapy with the anthracycline-based regimen R-CHOP was indicated for 51.9% (81/156), R-CVP was used in 23.7% (37/156), ICT based on purine analogs in 13.4% (21/156) and palliative protocols were applied in 10.9% (17/156) of cases. Almost 30% (46/156) of FL patients received four cycles and 52.5% (82/156) completed eight cycles of ICT. Maintenance with rituximab was not available in our institution and only 12.5% (20/160) of cases had access to this strategy in clinical trials. Over the whole follow-up, only 8.1% (13/160) of patients underwent autologous hematopoietic stem cell transplantation (ASCT).

The ORR with first-line therapy was 82.7% (129/156) (95% CI: 76.1–88.0%) for AS-HTB patients, 53.2% (83/156) (95% CI: 45.3–60.9%) obtained CR and 29.4% (46/156) (95% CI: 22.7–36.9%) achieved partial response (PR). Primary refractory disease, including stable disease and primary progressive disease after up-front therapy, was found in 12.1% (19/156) (95% CI: 6.32%–21.1%), and in 5.1% (8/156) of cases response criteria were not assessed. Considering the different regimens, the ORR for R-CHOP, R-CVP, ICT-P and palliative regimens were 87.7% (95% CI: 79.2–93.5%), 78.4% (95% CI: 63.3–89.2%), 90.5% (95% CI: 72.8–98.0%) and 58.8% (95% CI: 35.6–79.3%), respectively, *p* = 0.023.

With a median follow-up of 7.6 years (95% CI: 6.3–9.0), the median PFS was 5.29 years (95% CI: 4.43 not reached) and the median OS was 11.4 years (95%CI: 8.22 not reached). The estimated 5-year PFS and OS for AS-HTB cases were 52.9% (95% CI: 44.7–62.7%) and 70.6% (95% CI: 63.6–78.3%), respectively.

Sixty-eight of 156 AS-HTB FL patients (43.5%) (95% CI: 35–50.2%) presented progression/relapsed disease (PD) during the whole follow-up. Furthermore, POD-24 rate was 21.7% (34/156) (95% CI: 15.8–28.7). There were no statistically significant differences in POD-24 rates among patients receiving R-CHOP (18.5%, 95% CI: 11.4–27.0%), R-CVP (29.7%, 95% CI: 17.6–44.0%) and ICT based on purine analogs (9.5%, 95% CI: 3.3–24.6%). However, statistically significant differences were observed in POD-24 rates among palliative regimens (35.3%, 95% CI: 23.7–63.8%) and the other more intensive ICT schemes, *p* = 0.031, as displayed in Table 4.

The OMR was 38.7% (62/160) (95% CI: 31.4–46.4%) for this subgroup. The majority of deaths occurring under PD—19/62 (30.6%), and 27.4% (17/62) were caused by infection. OMR in AS-HTB patients treated with R-CHOP, R-CVP, ICT-P and palliative protocols were 33.3% (95% CI: 23.8–44.0%), 40.5% (95% CI: 25.9–56.6%), 33.3% (95% CI: 16.3–54.6%), and 64.7% (95% CI: 41.1–83.7%), respectively, *p* = 0.105.

The median OS for R-CHOP was not reached, and was 9.74 years (95 CI%: 8.69 not reached) for R-CVP; it was also not reached for ICT based on purine analogs at 5.73 years (95 CI%: 2.4 not reached) for palliative protocols. The estimated 5-year OS for R-CHOP, R-CVP, ICT based on purine analogs and palliative regimens not containing rituximab was 72.9% (95% CI: 63.3–83.8%), 69.4% (95% CI: 55.8–86.3%), 85.0% (95% CI: 70.7–100%) and 52.9% (95% CI: 33.8–82.9%), *p* = 0.130. The median PFS according to the treatment groups R-CHOP, R-CVP, and palliative regimens, in years, was 4.96 years (95% CI: 3.76 not reached), 5.29 (95% CI: 2.42 not reached), and 2.29 years (95% CI: 1.54 not reached), respectively. In the subgroup of patients treated with ICT based on purine analogs, median PFS was not reached. The estimated 5-year PFS for R-CHOP, R-CVP, ICT-P and palliative strategies was 49.8% (95% CI: 38.4–64.5%), 54.7% (95% CI: 39.5–79.9%), 72.2% (95% CI: 54.2–96.2%) and 33.8% (15.6–76.6%), respectively, *p* = 0.024 (Figure 4).

### 3.4. Histological Transformation and Event Adverse Profile

The cumulative annual risk for HT to HGBCL was 0.5%, with a median time since the diagnosis until HT of 15.6 years (95% CI: 14.7 not reached) for the whole cohort, i.e., 15.6 years (95% CI: 14.7 not reached) for ES disease, 14.7 years (95% CI: 11.8 not reached) for AS-HTB, and not reached for AS-LTB, *p* = 0.06. In the whole cohort, the HT rate was 13.5% (29/214) (95% CI: 9.46–18.42%), i.e., 8.0% (2/25) (95 CI%:1.7–23.2%) for ES FL patients, 6.9% (2/29) (95% CI:1.46–20.3%) for AS-LTB and 15.6% (25/160) (95% CI: 10.6–21.8%) for AS-HTB, *p* = 0.06. The cumulative rate of HT for the whole cohort for subgroups stratified by clinical staging and tumor burden is represented in Figure 5.

Concerning therapy toxicities, FL patients receiving ICT regimens in first or subsequent treatment lines developed neutropenia, thrombocytopenia and febrile neutropenia in 63.0% (135/214) (95 CI%: 56.7–69.6%), 23.9% (51/214) (95% CI: 18.5–30%) and 12.2% (26/214) (95% CI: 8.3–17.1%) of cases, respectively. Hospitalization related to treatment intolerance was necessary for 20.1% (42/214) (95 CI%: 15.1–25.9%) of patients, and blood transfusions were also necessary for 20.1% (42/214) (95 CI %: 15.1–25.9%) of cases throughout up-front ICT. Decision on therapy discontinuation was driven by intolerance, recurrent infection, lack of response, progressive disease or death. Considering the ICT regimens, there was no difference in rates of thrombocytopenia, treatment interruption, hospitalization and transfusion dependency between the four main treatment subgroups. On the other hand, patients receiving ICT based on anthracyclines or purine analogs developed higher rates of G3/G4 neutropenia than those treated with R-CVP (*p* < 0.001), but this was not translated into higher rates of febrile neutropenia (*p* = 0.301). The event adverse profile stratified by treatment subgroups is summarized in Table 5.

### 3.5. Prognostic Factors

#### 3.5.1. Univariate Analysis

In the univariate analysis, variables associated with decreased OS in the whole cohort of 214 FL patients were AS-HTB (HR: 2.70, 95% CI: 1.00–7.67, *p* = 0.047), loss of CD10 expression (HR: 1.90, 95% CI: 1.03–3.57, *p* = 0.039), Ki-67 ≥ 30% (HR: 1.90, 95% CI: 1.12–3.20, *p* = 0.016), ECOG ≥ 2 (HR: 3.67, 95% CI: 2.19–6.12, *p* < 0.001), ≥2 comorbidities (HR: 1.97, 95% CI: 1.09–3.54, *p* = 0.024), hemoglobin < 120 g/L (HR: 2.43, 95% CI: 1.52–3.90, *p* < 0.001), β2M ≥ 1.7 mg/dL (HR: 2.80, 95% CI: 1.74–4.48, *p* < 0.001), serum albumin < 3.5 g/dL (HR: 4.58, 95% CI: 2.57–8.17, *p* < 0.001), globulin ≤ 1.5 g/dL (HR: 4.71, 95% CI: 1.70–13.0, *p* = 0.003), B-symptoms (HR: 3.30, 95% CI: 1.96–5.54, *p* < 0.001), bulky ≥ 7 cm (HR: 2.00, 95% CI: 1.28–3.30, *p* = 0.003), extra nodal involvement (HR: 1.91, 95% CI: 1.09–3.34, *p* = 0.022), high-risk FLIPI (HR: 7.00, 95% CI: 2.55–19.50, *p* < 0.001) and HT to HGBCL (HR: 1.87, 95% CI: 1.08–3.23, *p* = 0.024)

Variables associated with decreased PFS were high LDH levels (HR: 1.39, 95% CI: 1.05–1.84, *p* = 0.021), globulin ≤ 1.5 g/dL (HR: 3.40, 95% CI: 1.06–10.8, *p* = 0.039), AS-HTB disease (HR: 2.00, 95% CI: 0.99–4.26, *p* = 0.05), B-symptoms (HR: 2.72, 95% CI: 1.76–4.22, *p* < 0.001), involvement of ≥4 nodal areas (HR: 1.08, 95% CI: 0.99–1.17, *p* = 0.065), ITP (HR: 2.08, 95% CI: 1.00–4.32, *p* = 0.048), BM infiltration (HR: 1.45, 95% CI: 0.95–2.20, *p* = 0.08), high-risk FLIPI (HR: 1.43, 95% CI: 1.09–1.88, *p* = 0.009), high-risk FLIPI-2 (HR: 1.39, 95% CI: 1.03–1.88, *p* = 0.029), and HT to HGBCL (HR: 3.91, 95% CI: 2.43–6.30, *p* < 0.001). Maintenance therapy with rituximab at first-line was associated with better PFS (HR: 0.31, 95% CI: 0.14–0.67, *p* = 0.003).

Predictors associated with higher risk for HT to HGBCL included loss of BCL-2 expression (HR: 1.67, 95% CI: 0.95–2.63, *p* = 0.071), ECOG ≥ 2 (HR: 2.99, 95% CI: 1.26–7.11, *p* = 0.013), elevated patient LDH/LDH upper limit ratio (HR: 1.69, 95% CI: 1.15–2.48, *p* = 0.007), β2M ≥ 1.7 mg/dL (HR: 2.42, 95% CI: 1.12–5.20, *p* = 0.024), B-symptoms (HR: 3.05, 95% CI: 1.39–6.68, *p* = 0.005), bulky ≥ 7 cm (HR: 2.51, 95% CI: 1.15–5.48, *p* = 0.02), involvement of ≥4 nodal areas (HR: 1.20, 95% CI: 1.04–1.39, *p* = 0.011), extra nodal disease (HR: 2.95, 95% CI: 1.12–7.79, *p* = 0.029), BM infiltration (HR: 2.29, 95% CI: 1.03–5.05, *p* = 0.04), high-risk FLIPI (HR: 1.89, 95% CI: 1.12–3.17, *p* = 0.016), high-risk FLIPI-2 (HR: 1.92, 95% CI: 1.09–3.37, *p* = 0.023), vital organ compression (HR: 2.62, 95% CI: 1.26–5.46, *p* = 0.010), thrombocytopenia < 150 × 10^9^/L (HR: 2.43, 95% CI: 1.16–5.07, *p* = 0.017), febrile neutropenia during primary therapy (HR: 3.58, 95% CI: 1.56–8.21, *p* = 0.003), POD-24 (HR: 4.42, 95% CI: 2.04–9.59, *p* < 0.001). Albumin ≥ 3.5 g/L (HR: 0.189, 95% CI: 0.07–0.47, *p* < 0.001) was associated with lower risk for HT. Figure 6 summarizes the main variables identified in univariate analysis able to predict OS, PFS and HT to HGBCL.

In univariate analysis, and considering only patients with AS-HTB, higher risk for POD-24 occurrence was detected in individuals presenting high-risk FLIPI-2 (HR: 3.26, 95% CI: 0.98–10.75, *p* = 0.05), involvement of ≥3 nodal sites > 3 cm (HR: 1.87, 95% CI: 1.02–3.44, *p* = 0.041) and treatment with palliative regimens not containing rituximab (HR: 3.18, 95% CI: 1.41–7.14, *p* = 0.041). Maintenance therapy after induction ICT was associated with lower POD-24 rates (HR: 0.209, 95% CI: 0.05–0.86, *p* = 0.03).

#### 3.5.2. Multivariate Analysis

In multivariate analysis the predictors associated with decreased OS were ECOG ≥ 2 (HR: 2.03, 95% CI: 1.16–3.56, *p* = 0.013), ≥2 comorbidities (HR: 2.10, 95% CI: 1.13–3.90, *p* = 0.018), hypoalbuminemia < 3.5 g/dL (HR: 2.88, 95% CI: 1.53–5.42, *p* < 0.001) and B- symptoms (HR: 2.50, 95% CI: 1.45–4.30, *p* < 0.001); poor PFS was independently associated with B-symptoms (HR: 2.50, 95% CI: 1.60–3.91, *p* < 0.001) and HT (HR: 3.57, 95% CI: 2.19–5.84, *p* < 0.001). Higher risk of POD-24 was associated with grade 3A FL status (HR: 3.70, 95% CI: 1.30–10.5, *p* = 0.014), ≥3 nodal areas > 3 cm (HR: 2.08, 95% CI: 1.10–3.03, *p* = 0.003) and monotherapy regimens not-containing rituximab (HR: 3.54, 95% CI: 1.56–8.09, *p* = 0.003). Clinical predictors independently associated with higher risk of HT were hypoalbuminemia (HR: 4.22, 95% CI: 1.50–11.81, *p* = 0.006) and organ-damage compression (HR: 3.10, 95% CI: 1.41–6.79, *p* = 0.005). Table 6 summarizes the results of the multivariate analysis, comprising prognostic factors independently associated with decreased OS, PFS, as well as increased risk for POD-24 and HT to HGBCL. Results are presented in HR, 95% CI, and *p*-value.

## 4. Discussion

In this study, we reported real-world data from a cohort of FL patients followed at a large referral center for cancer treatment in Brazil over more than two decades. In agreement with the recommendations of the current medical literature, the diagnosis of FL in our cohort was performed through lymph node biopsy in most cases. As expected, CD10 and BCL-2 positivity in tumor samples were present in most low-grade cases, as in those with predominance of histologic grade 1/2 [53]. However, although most cases presented lymph node enlargement at diagnosis, and in agreement with previous studies, more than half of the cases demonstrated extra nodal involvement by lymphoma, especially bone marrow infiltration. Gogia et al., reported extra nodal involvement in FL ranging from 22% to 46%, but other cohorts showed higher frequencies of bone marrow involvement, reaching up to 70% of cases [54]. Other frequent sites of extra nodal involvement in our cohort were pleural and peritoneal cavities (21.5%). Although GELF criteria consider pleural and peritoneal involvement to indicate treatment, there is not much data assessing its occurrence in FL. Morel et al., in a report on 91 patients with FL, found 14.5% pleural effusion, and this clinical feature was associated with poor OS [55]. In agreement with these reports, we demonstrated that serous effusions and extra nodal involvement predicted decreased OS in our cohort.

In opposition to previous studies, we found a higher number (44.3%) of patients with B- symptoms, and this finding was independently associated with poor OS and PFS in our cohort. Jacobsen and Freedman A. reported that B-symptoms are usually found in up to 20% of FL [56] and, according to data from the Prima study, which included only advanced-stage and high tumor burden FL, only approximately o30% of patients presented B-symptoms [57]. We interrogate if this higher frequency of B-symptoms in our patients could be explained by the fact that, in Brazil, as well as in other low- and middle-income countries, there is substantial delay from the beginning of symptoms until to the definitive diagnosis of lymphoma, which adversely impacts the prognosis of individuals with different NHLs subtypes, as previously demonstrated by our group [58].

As expected, most of our patients showed advanced-stage disease (Ann Arbor III and IV) at diagnosis and 74.7% (160/214) presented one or more GELF criteria, indicating high tumor burden or symptomatic neoplasm, and leading to the need to initiate immunochemotherapy [29]. However, bulky disease ≥ 7 cm was more frequent in our cohort (44.3%) than is usually reported in the literature. In this sense, Gogia et al., described only 19% bulky disease in a cohort of 181 FL patients [54]. In addition, high-risk FLIPI and FLIPI-2 scores were found in most cases of our cohort, i.e., 49.5% and 44.8%, respectively, and intermediate- and high-risk FLIPI/FLIPI-2 comprised 77.5% and 84.9% of patients, respectively. These findings show that our cohort presented clinical features associated with dismal prognosis, reflecting an unselected and typical real-world sample.

As described in recent studies, in our cohort, the risk of HT to HGBCL was 0.5% per year. With a median time from diagnosis to HT of 15.6 years, we observed a 13.5% (29/214) rate of HT into HGBCL. Similar to our data, most studies assessing HT rate were based oon retrospective analysis. In fact, prior to the rituximab era, the estimated risk for HT in FL patients ranged from 24% to 70% [59,60], from 11% to 17% at 5 years [59,61] and 30% at 10 years [62]. These different HT rates can probably be explained due to factors such as cohort size, different criteria used to characterize histological transformation, time of follow-up, and obtaining a new biopsy at the time of disease relapse or progression [60,63,64]

However, more recent studies incorporating rituximab use in up-front therapy settings demonstrated an HT risk of 2% to 3% 10–15 years from FL diagnosis. Al-Tourah AJ et al. analyzed a large cohort of FL in Canada and showed that the HT risk remained stable over time, achieving a cumulative HT rate of 3% per year [65]. Similarly, in a cohort composed of 325 FL patients followed for 16 years in the United Kingdom, the HT rate was 2% to 3% per year [62]. Even though these results have been confirmed by other authors in the immunochemotherapy era [61,66,67] there is no consensus around the risk factors associated with HT to HGBCL [68]. In our study, in multivariate analysis, the predictive factors associated with higher risk of HT were hypoalbuminemia (serum albumin < 3.5 g/dL) and organ-damage compression by lymphoma. In addition, FL patients categorized as AS-HTB presented higher rates of HT in comparison to ES and AS-LTB, *p* = 0.06.

As described by Casulo et al., in our cohort, POD-24 was found in 21.7% of cases [33]. In multivariate analysis, the presence of three or more nodal lesions ≥ 3 cm was associated with increased risk of POD-24, as well as high content of centro-blasts (histologic grade 3A) and up-front regimens lacking an anti-CD20 monoclonal antibody. A meta-analysis carried out on 31 studies was recently conducted aiming to establish clinic-laboratorial predictors for POD-24 in FL patients. In this study, Gao et al. identified 11 risk factors for POD-24, including elevated sIL-2R, β2M, LDH, total metabolic volume > 510 cm^3^, vitamin D < 20 ng/mL, grade 3A and lymphoma containing ≥ 15 macrophages/HPF [69]. However, to date there is no consensus on the identification of robust predictive factors for POD-24 across different studies. Interestingly, Freeman et al., demonstrated that around 75% of FL patients developing POD-24 experienced it in the context of HT for aggressive histology at the moment of progression [70]. Considering that, some authors emphasize that a new biopsy should be routinely performed in patients with POD-24 [71], in our institution we usually indicate a new biopsy for all FL patients who had PD or relapse, particularly in POD-24 context, to rule out HT to HGBCL. Contrary to what was expected, we did not find any association between high-risk categories of FLIPI and FLIPI-2 scores and higher POD-24 rates.

As expected, FL patients diagnosed at ES and AS-LTB showed increased OS in comparison to patients with AS-HTB, *p* = 0.006. However, there was no difference in PFS among these groups, *p* = 0.260. Our FL patients with ES were kept in WW strategy or treated with IF-RT at low doses. The ORR of those treated with IF-RT was 100%, with high rates of CR (83.3%). The median PFS and OS for the whole cohort of ES disease was 9.37 years and not reached, respectively. Typically, ES FL is treated with external beam radiotherapy with or without systemic therapy, presenting excellent disease control and long-term complete remissions in more than half of cases [72,73]. Similarly, and in agreement with current recommendations, in our cohort most FL patients categorized as AS-LTB were kept in WW strategy and presented prolonged PFS and OS. From this perspective, with a median follow-up of 10.3 years, the median PFS and OS for AS-LTB FL cases were 10.57 years and not reached, respectively [74].

As predicted, in our cohort, AS-HTB FL patients up-front treated with ICT regimens, such as R-CHOP, R-CVP or R-FC(M), had higher ORR than those using regimens based only in chemotherapy, *p* = 0.023. Similarly, the overall mortality rate was higher in patients receiving chemotherapy in comparison to ICT, *p* = 0.031. In fact, the incorporation of immunotherapy with anti-CD20 rituximab was a hallmark of FL therapy. The use of ICT in first-line increased both OS and PFS in comparison to chemotherapy alone [75,76,77,78,79]. However, maintenance with immunotherapy after up-front therapy showed increment only in PFS, but not in OS [80].

Although the most widely used front-line regimens for FL are rituximab plus bendamustine and R-CHOP [81], other regimens, such as R-CVP [77,82,83], are also adopted, despite offering lower response rates, particularly for frail and elderly patients. In our institution, as bendamustine is not available, we routinely use R-CHOP and R-CVP. Over this long period of 22 years, treatment in our institution changed according to the new data provided by clinical trials, and depended on the incorporation of new drugs, especially those of high cost, such as anti-CD20 monoclonal antibody, provided by the Brazilian public health system. In fact, although, the Food and Drug Administration [84] approved rituximab for FL in 1997 and in 1998 via the Brazilian Health Regulatory Agency (ANVISA), this drug was only available in public health care for FL in our country in 2014. Therefore, in our cohort there were some FL patients treated with no immunotherapy. These cases, basically treated with monotherapy containing chlorambucil or oral fludarabine, were considered as having received a palliative approach. In addition, few cases were treated with immunotherapy associated with purine analogous [FCR or FCR(M)] regimens, especially those diagnosed a long time ago. In the timeline of our institution, we used R-CVP for a while, and more recently adopted R-CHOP as the standard regimen for FL patients presenting symptomatic disease or high tumor burden according to GELF criteria. Our results showed similar ORR with R-CHOP, R-CVP and R-plus purine analogs, and lower rates when isolated chemotherapy was compared with the three previously mentioned ICT regimens. The median PFS and OS, as well as the estimated 5-year PFS and OS, were not distinct in these different ICT protocols, but was also lower for patients treated with CT alone.

Finally, according to our results obtained in multivariate analysis, ECOG ≥ 2, ≥2 comorbidities, hypoalbuminemia < 3.5 g/dL and B-symptoms were independently associated with decreased OS, and B-symptoms and HT to HGBCL were associated with poor PFS. In addition, FLIPI and FLIPI-2 scores failed to demonstrate impact on OS and PFS in this real-world FL cohort.

## 5. Conclusions

In conclusion, we demonstrated based on real-world evidence that FL is a malignancy characterized by marked clinical–prognostic heterogeneity, translated into diverse clinical staging and tumor burden subcategories. Here, we also showed that FL patients classified as AS-HTB presented decreased survival and higher rates of HT to HGBCL compared to ES and AS-LTB cases. Furthermore, poor performance status (ECOG ≥ 2) and presence of B-symptoms, hypoalbuminemia, and more than 2 comorbidities were independently identified as predictors for decreased overall survival in patients with cFL. Additionally, the prognostic factors identified in our analysis may help to identify FL patients with higher-risk of HT and early-progression (POD-24), helping clinicians to identify high-risk patients and establish personalized treatment strategies.

## Figures and Tables

**Figure 1 cancers-16-03914-f001:**
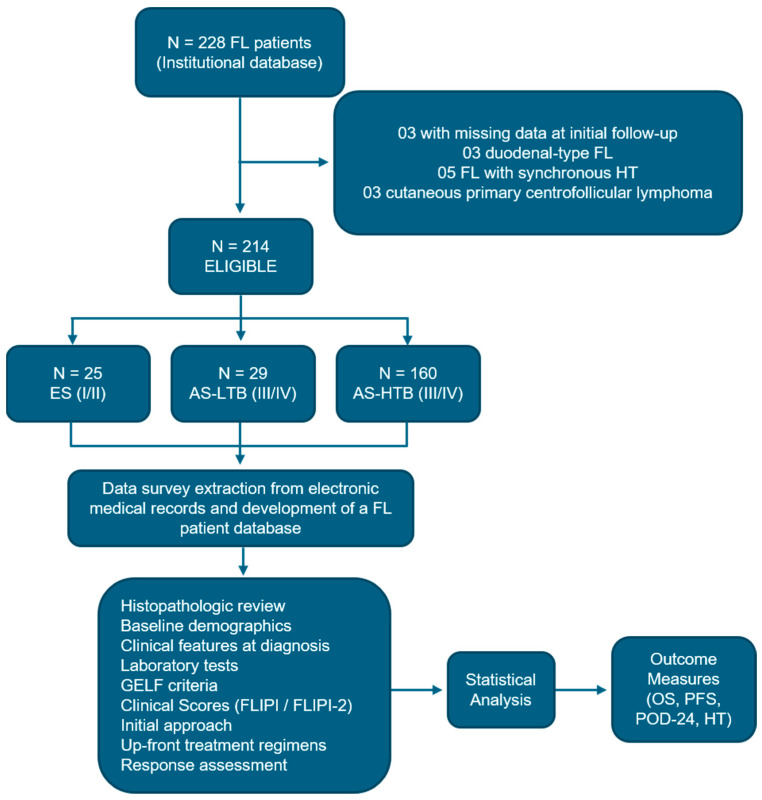
Study design.

**Figure 2 cancers-16-03914-f002:**
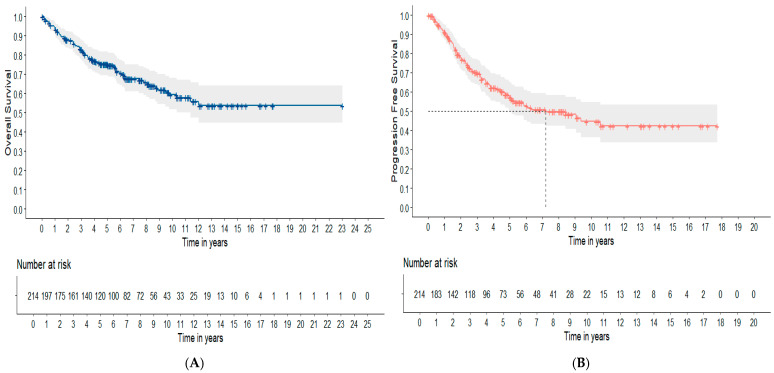
OS (**A**) and PFS (**B**) in the whole cohort of FL patients grade 1–3A (N = 214). The survival curves were constructed using the Kaplan–Meier method.

**Figure 3 cancers-16-03914-f003:**
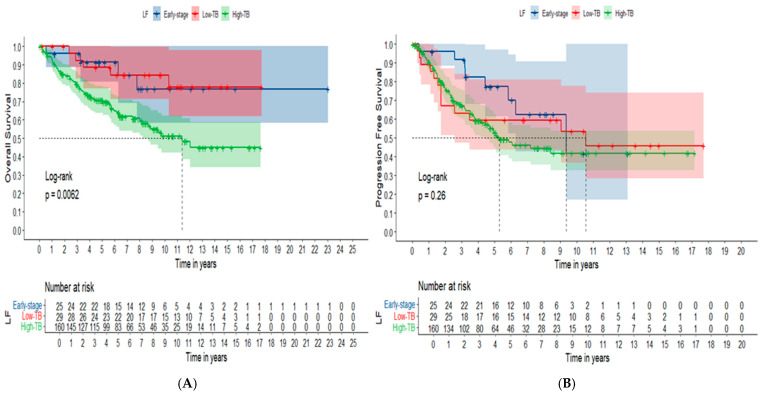
OS (**A**) and PFS (**B**) curves according to subgroup stratified by clinical staging and tumor burden (N = 214). Blue-line: ES; Red-line: AS-LTB; Green-line: AS-HTB. The survival curves were constructed using the Kaplan–Meier method. The comparison among curves used the Log-Rank test.

**Figure 4 cancers-16-03914-f004:**
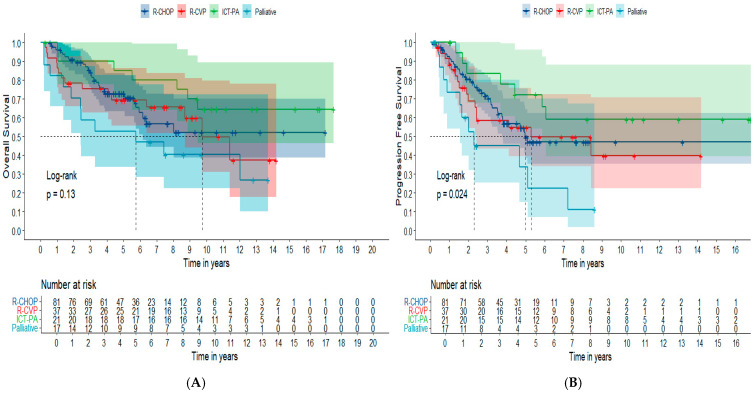
OS curves (**A**) and PFS (**B**) for AS-HTB FL patients according to up-front therapy (N = 156). Dark-blue line: R-CHOP; Red-line: R-CVP; Green-line: ICT-P; Light-blue line: palliative regimens. The survival curves were constructed using the Kaplan–Meier method. The comparison among curves used the Log-Rank test.

**Figure 5 cancers-16-03914-f005:**
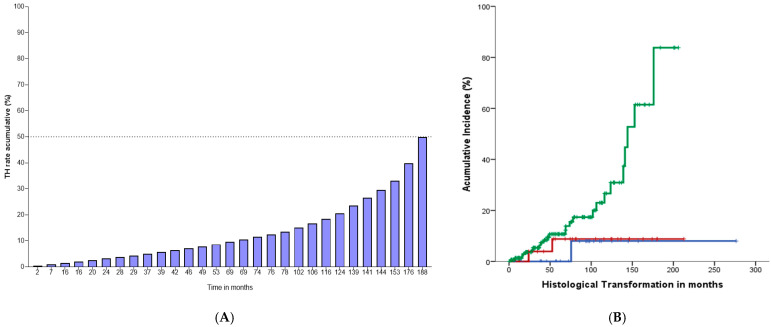
Annual cumulative rate for HT to HGBCL in the whole cohort (**A**) and cumulative rate for HT to HGCBL according to clinical staging and tumor burden (**B**). Blue-line: ES; Red-line: AS-LTB; Green-line: AS-HTB, *p* = 0.06. The comparison of cumulative incidence for HT to HGBCL among different categories was calculated using the Log-Rank test.

**Figure 6 cancers-16-03914-f006:**
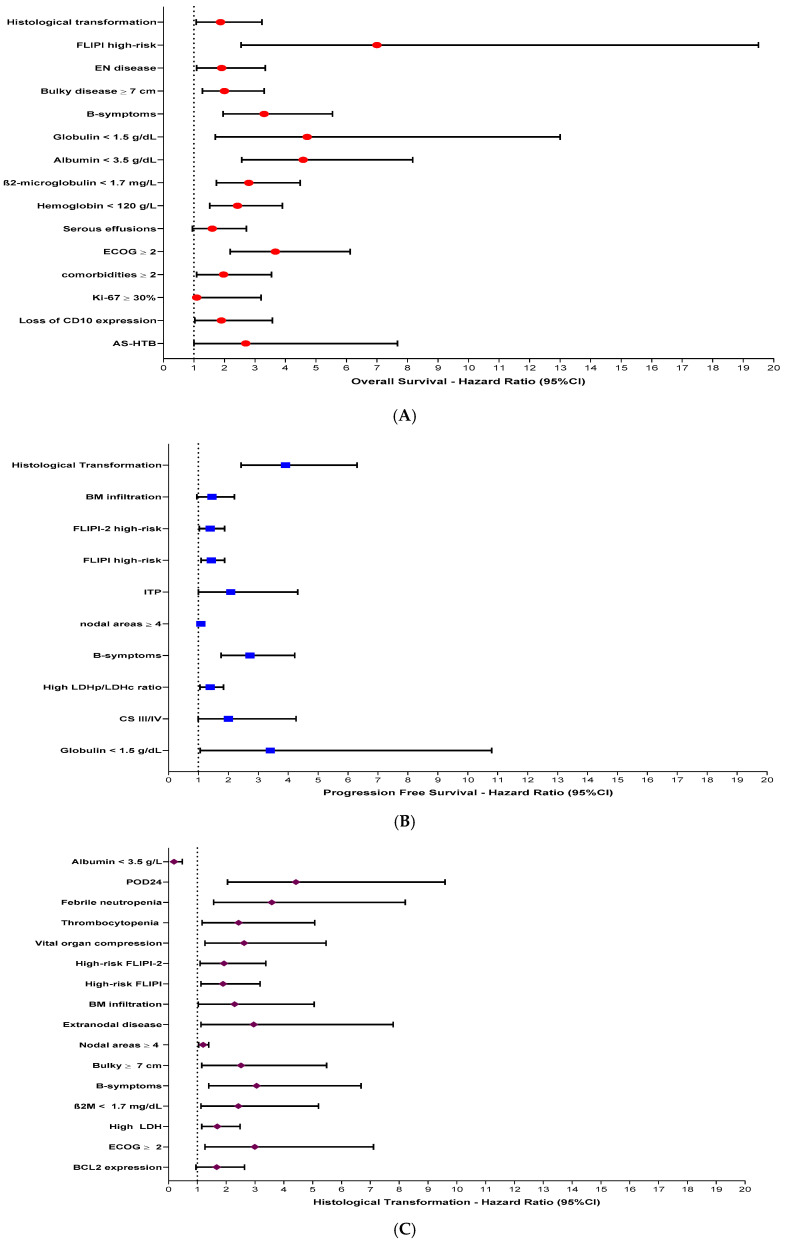
Prognostic factors for OS (**A**), PFS (**B**) and HT to HGBCL (**C**) identified in univariate analysis. Univariate analysis used the Cox regression method and the results were presented as forest plots.

**Table 1 cancers-16-03914-t001:** Up-front therapy regimens adopted in the study population.

Therapy Regimens	Protocols
Observation	Watchful and waiting
R-CHOP every 21 days	D1: Rituximab 375 mg/sqm I.V.; Cyclophosphamide 750 mg/sqm I.V.; Doxorubicin 50 mg/sqm I.V.; Vincristine 1.4 mg/sqm (max. 2.0 mg) I.V.; D1–5: Prednisone 100 mg/d P.O.; up to 6–8 cycles
R-CVP every 21 days	D1: Rituximab 375 mg/sqm I.V.; Cyclophosphamide 750 mg/sqm I.V.; Vincristine 1.4 mg/sqm (max. 2.0 mg) I.V.; D1–5: Prednisone 100 mg/d. P.O.; up to 8 cycles
ICT based on purine analogs	R-FC (Rituximab 375 mg/sqm I.V. on D1; Fludarabine 25 mg/sqm I.V. on D1–D3; Cyclophosphamide 250 mg/sqm I.V. on D1–D3) every 28 days; up to 6 cycles R-FCM (Rituximab 375 mg/sqm I.V. on D1; Fludarabine 25 mg/sqm I.V. on D1–D3, Cyclophosphamide 250 mg/sqm I.V. on D1–3, and Mitoxantrone 8 mg/sqm I.V. on D1, every 28 days); up to 6 cycles R-F (Rituximab 375 mg/sqm I.V. on D1 plus fludarabine 40 mg/sqm P.O. on D1–D5, every 28 days); up to 6 cycles
Palliative strategies	Chlorambucil 10 mg/sqm P.O. on D1–D6 28 day-cycles; up to 8–12 cycles; Fludarabine 40 mg/sqm P.O.on D1–D5 28 day cycle; up to 6 cycles.

**Table 2 cancers-16-03914-t002:** Baseline demographic and clinical characteristics of FL grades 1–3A (N = 214).

Clinical Characteristics	Follicular Lymphoma (N = 214)	(%)
Female	119	55.6
Age (median) years	60 (IQR 25–75%: 51–69)	-
≥2 comorbidities	29	13.5
ECOG ≥ 2	33	15.4
Stage I/II Stage III/IV	29 185	13.5 86.4
B-symptoms	106	49.5
Bulky ≥ 7 cm	95	44.3
Serous effusion	46	21.5
Circulating B neoplastic cells	25	11.7
Liver/splenic involvement	30	14
Extra nodal disease	142	66.3
Bone marrow involvement	106	49.5
≥2 extra nodal areas	91	42.5
Organ compression symptoms	52	24.3
>4 nodal areas	147	68.6
≥3 nodal areas > 3 cm	64	29.9
FLIPI score -low/-interm/-high risk	48/60/106	22.4/28/49.5
FLIPI-2 score -low/-interm/-high risk	32/86/96	14.9/40.1/44.8
Lymphocytosis > 5 × 10^9^/L	22	10.3
Cytopenias	18	8.4

IQR: Interquartile range; ECOG: Eastern Cooperative Oncology Group; FLIPI: Follicular Lymphoma International Prognostic Index; FLIPI-2: Follicular Lymphoma International Prognostic Index 2.

**Table 3 cancers-16-03914-t003:** Clinical and laboratory findings according to clinical staging and tumor burden status.

Baseline Clinical Characteristics	ES (I/II) 11.6% (25/214)	AS (III/IV)-LTB 13.5% (29/214)	AS (III/IV)-HTB 74.7% (160/214)	*p*-Value
Female	48% (12/25)	62% (18/29)	55.6% (89/160)	0.584
Age > 60 years	56% (14/25)	44.8% (13/29)	45% (72/160)	0.584
ECOG ≥ 2	4.0% (1/25)	0% (0/29)	20% (32/160)	0.006
≥2 comorbidities	12% (3/25)	6.9% (2/29)	15% (24/160)	0.488
Serous effusions	0% (0/25)	0% (0/29)	27.5% (44/160)	<0.001
Leukemic phase	0% (0/25)	3.4% (1/29) #	15% (24/160)	0.031
Hepato/Splenomegaly	0% (0/25)	0% (0/29)	18.7% (30/160)	<0.001
Involvement of vital organs	0% (0/25)	0% (0/29)	13.1% (21/160)	<0.001
Organ damage/compression	0% (0/25)	0% (0/29)	32.5% (52/160)	<0.001
B-symptoms	0% (0/25)	0% (0/29)	66.2% (106/160)	<0.001
Bulky ≥ 7 cm	4% (1/25)	0% (0/29)	58.7% (94/160)	<0.001
≥3 nodal areas > 3 cm	0% (0/25)	0% (0/25)	40% (64/160)	<0.001
≥4 nodal areas	12% (3/25)	62% (18/29)	78.7% (126/160)	<0.001
EN disease	12% (2/25) *	48.2% (14/29)	78.1% (125/160)	<0.001
BM involvement	0% (0/25)	37.9% (11/29)	59.3% (95/160)	<0.001
FLIPI score - Low - Intermediate - High	80% (20/25) 16% (4/25) 4% (1/25)	34.4% (10/29) 41.3% (12/29) 24.1% (7/29)	11.2% (18/160) 27.5% (44/160) 61.2% (98/160)	<0.001
FLIPI-2 score - Low - Intermediate - High	56% (14/25) 44% (11/25) 0% (0/25)	27.5% (8/29) 58.6% (17/29) 13.7% (4/29)	6.2% (10/160) 36.2% (58/160) 57.5% (92/160)	<0.001
Anemia	12.5% (1/25) **	6.9% (2/29)	35.0% (56/160)	<0.001
WBC > 11 × 10^9^/L	8% (2/25)	6.9% (2/29)	16.2% (26/160)	0.064
>5 × 10^9^/L total lymphocytes	0% (0/25)	3.4% (1/29) #	18.7% (30/160)	0.002
Monocytosis	4% (1/25)	0% (0/29)	14.3% (23/160)	0.069
Thrombocytopenia	0% (0/25) **	0% (0/29) **	24.3% (39/160)	0.205
Hypoalbuminemia	0% (0/25)	0% (0/29)	13.1% (21/160)	0.020
Paraproteinemia	8% (2/25)	6.9% (2/29)	8.1% (13/160)	0.975
AIHA	0% (0/25)	0% (0/29)	0.63% (1/160)	0.844
ITP	0% (0/25)	6.9% (2/29)	7.5% (12/160)	0.369

ES: early-stage; AS (III/IV)-LTB: advanced stage disease with low tumor burden/asymptomatic; AS III/IV)-HTB: advanced stage disease with high tumor burden/symptomatic; ECOG: *Eastern Cooperative Oncology Group*; EN: extra nodal; BM: bone marrow; FLIPI: *Follicular Lymphoma International Prognostic Index*; FLIPI-2: *Follicular Lymphoma International Prognostic Index 2*; Anemia: hemoglobin < 120 g/L; WBC: white blood cell count; monocytosis > 1 × 10^9^/L; thrombocytopenia < 100 × 10^9^/L; hypoalbuminemia: albumin < 3.5 g/dL; AIHA: autoimmune hemolytic anemia; ITP: immune thrombocytopenia. * Two patients had localized extra nodal asymptomatic disease, 1 non-Bulky and 1 bulky mass and were classified as ES. ** one patient in ES had mild anemia (Hb 117 g/L), mild thrombocytopenia: 2 in ES and 6 in AS-LTB (100–150 × 10^9^/L) at diagnosis that were not related to FL. # One patient presented >5.0 × 10^9^/L total lymphocytes, but less than 5.0 × 10^9^/L clonal lymphocytes in PB, being categorized as AS-LTB. The variables were compared using the Chi-square test and/or the Fisher’s exact test.

**Table 4 cancers-16-03914-t004:** Outcomes in the whole cohort (N = 214) and by subgroups (stage/tumor burden) of FL.

Outcome	Total Cohort Absolute (%)	ES I/II (N = 25)	AS-LTB (III/IV) (N = 29)	AS-HTB (III/IV) (N = 160)	*p*-Value
Death	71 (33%)	4 (16%)	5 (17.2%)	62/160 (38.7%)	0.012
PD/Relapse	89 (41.5%)	8 (32%)	13 (44.8%)	68/156 (43.5%)	0.569
POD-24	34 (15.8%)	- - -	- - -	34/156 (21.7%)	- - -
Transformation to high-grade B cell lymphoma	29 (13.5%)	2 (8%)	2 (6.9%)	25/160 (15.6%)	0.060
**POD-24 (AS-HTB)** **N (%)**	**R-CHOP** **(N = 81)**	**R-CVP** **(N = 37)**	**ICT-P** **(N = 21)**	**Palliative** **(N = 17)**	- - -
34 (21.7%)	15 (18.5%)	11 (29.7%)	2 (9.5%)	6 (35.3%)	0.031

ES: early stage; AS-LTB: advanced stage low tumor-burden; AS-HTB: advanced stage high tumor-burden; PD: progression of disease; POD-24: progression of disease within 24 months from first-line treatment; R: rituximab; CHOP: cyclophosphamide, doxorubicin, vincristine and prednisone; CVP: cyclophosphamide, doxorubicin, vincristine and prednisone; ICT-P: immunochemotherapy based on purine analogs. All rates were compared using the Chi-square test and/or the Fisher’s exact test.

**Table 5 cancers-16-03914-t005:** Toxicity profile according to up-front or subsequent ** ICT regimens adopted in the FL cohort (** FL patients firstly managed with IF-RT, rituximab monotherapy and WW strategy subsequently exposed to ICT).

	R-CHOP N = 93	R-CVP N = 44	ICT-PA N = 26	Palliative Regimens N = 21	*p*-Value
G3/G4 Neutropenia	81.7% (72.9–88.5%) N = 76	47.7% (33.5–62.2%) N = 21	80.7% (62.8–92.2%) N = 21	76.1% (55.4–90.2%) N = 16	*p* < 0.001
G3/G4 Febrile neutropenia	15.0% (8.9–23.3%) N = 14	6.8% (1.9–17.0%) N = 3	15.3% (5.4–32.5%) N = 4	23.8% (9.7–44.5%) N = 5	*p* = 0.301
Ge/G4 Thrombocytopenia	24.7% (16.8–34.1%) N = 23	25.0 (14.0–39.1%) N = 11	42.3% (24.9–61.2%) N = 11	28.5%(12.9–49.7%) N = 6	*p* = 0.341
Treatment interruption	6.4% (2.7–12.8%) N = 6	11.3% (4.4–23.1%) N = 5	7.6% (1.6–22.4%) N = 2	19.0% (6.7–39.1%) N = 4	*p* = 0.310
Hospitalization	21.9% (14.4–31.2%) N = 20	18.6% (9.2–32.0%) N = 8	15.3% (5.4–32.5%) N = 4	40.0% (21.0–61.6%) N = 8	*p* = 0.196
Blood transfusion dependency	25.2% (17.2–34.8%) N = 23	13.9% (6.0–26.5%) N = 6	19.2% (7.7–37.1%) N = 5	35.0% (17.2–56.7%) N = 7	*p* = 0.250

G3 and G4 toxicities were considered according to CTCAE (*Common Terminology of Clinical Adverse Events*) v5.0; ICT-PA: immunochemotherapy regimens based on purine analogs. All variables were compared using the Chi-square test and/or the Fisher’s exact test.

**Table 6 cancers-16-03914-t006:** OS, PFS, POD-24 and HT predictors by multivariate analysis for the entire FL cohort.

Variable	OS [HR; 95% CI; *p*-Value]	PFS [HR; 95% CI; *p*-Value]	POD-24 [HR; 95% CI; *p*-Value]	HT [HR; 95% CI; *p*-Value]
ECOG ≥ 2	2.03; 1.16–3.56; 0.013			
≥2 comorbidities	2.10; 1.13–3.90; 0.018			
Albumin < 3.5 g/dL	2.88; 1.53–5.42; <0.001			4.22; 1.50–11.81; 0.006
B-Symptoms	2.50; 1.45–4.30; <0.001	2.50; 1.60–3.91; <0.001		
Organ compression				3.10; 1.41–6.79; 0.005
HT to HGBCL		3.57; 2.19–5.84; <0.001		------
≥3 nodal areas ≥ 3 cm			2.08; 1.10–3.93; 0.003	
Histological grade 3A			3.7; 1.30–10.5; 0.014	
CT not containing ritux.			3.54; 1.56–8.09; 0.003	

OS: overall survival; PFS: progression-free survival; POD-24: progression of disease within 24 months from up-front therapy; HT: histologic transformation; HGBCL: high-grade B-cell lymphoma; ECOG: *Eastern Cooperative Oncology Group*; CT: chemotherapy. Multivariate analysis used the Cox regression method.

## Data Availability

The data that support the findings of this study are available from the corresponding author upon reasonable request.

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
