# Peer review of "Clinical-Demographic Profile, Prognostic Factors and Outcomes in Classic Follicular Lymphoma Stratified by Staging and Tumor Burden: Real-World Evidence from a Large Latin American Cohort"

_cancers, 2024, doi:10.3390/cancers16233914_

Round 1
Reviewer 1 Report
Comments and Suggestions for Authors
Line 73-76: this sentence need more accurate classification Per WHO
Authors: Currently, the 5th. edition of the World Health Organization classification for Hematopoeitic and Lymphoid Tissue Tumors (WHO-HAEM 5th) recognizes four main subtypes of FL, including the classic follicular lymphoma (cFL), in situ FL, duodenal-type FL, and pediatric FL [1].
Recommendation: Currently, the 5th. edition of the World Health Organization classification for Hematopoietic and Lymphoid Tissue Tumors (WHO-HAEM 5th) recognizes four main types under FL, including follicular lymphoma (FL), in situ follicular B cell neoplasm, duodenal-type FL, and pediatric-type FL [1]. Additionally, The WHO consider four subtypes of FL as Classic FL (cFL); FL with unusual cytological features (uFL); FL with a predominantly diffuse growth pattern (dFL); follicular large B-cell lymphoma (FLBCL)
Line 82-83
Authors: new subtypes of FL have been recognized, including lymphoma BCL-2-rearranged-negative (BCL-2-R)
Recommendation: remove the word "new", these are not new entities and have been recognized for some time.
Recommendation: change the abbreviation as this abbreviation is very similar to the actual BCL2 rearranged FL used by the WHO"FL-BCL2-R" use for example "BCL2-R-negative" or "BCL2-non-R"
line 91 Nowadays, it was understood
Recommendation: change "was" to "is"
line 98
Authors: Additionally, the WHO 5th edition also recommended the name of FL with uncommon features for morphological variants, such as those characterized by large centrocytes or blastoid, and for those with a predominantly diffuse growth pattern [1].
Recommendation: please use the exact wording of the WHO "FL with unusual cytological features for morphological variants, such as those characterized by large centrocytes or blastoid morphology and FL with a predominantly diffuse growth pattern when the growth pattern is predominantly diffuse rather than follicular/nodular.
line 108
BCL2 gene rearrangement
line 108-109-110
Authors: It can be also justified by mutations in BCL-2 gene hindering the recognition by the monoclonal antibodies conventionally used for routine diagnosis by immunohistochemistry
Recommendation: The sentence is not clear, suggest rephrasing! do you mean "certain BCL2 gene mutations hinder recognition by the monoclonal antibodies immunostains used for routine staining "
line 123-124
Author: with no bulky disease ≥ 7 cm (early-stage/ES)
Recommendation: Do you mean < less than ?? as bulky disease is defined as 7 cm or more
line 150
Authors: Based in
Recommendation: based on
line 233-235
recommendation: remove it and rephrase too "When indicated, especially in patients who are candidate to regimens containing anthracycline transthoracic echocardiogram was also performed."
line 283
recommendation : every 6 months
line 285
recommendation : every 6 months
line 288-290
Authors: HT was defined by the presence of diffuse architecture of the compromised tissue associated with ≥ 20% of lymphoid large cells and/or Ki-67 index ≥ 50%[53].
Recommedation: The sentence is not clear as to the pathologic accuracy of this definition, the authors need to use a definition used by a majority of pathologist which is based on the WHO and make it clearer as "HT was defined by the presence of FL 3B or Diffuse large B cell lymphoma characterized by diffuse sheets of large cells. The Ki-67 proliferation is significant and is a prognostic indicator however it should not be used as a feature of transformation to large cell lymphoma.
Recommendation: isolate those patient with high Ki-67 proliferation and evaluate their clinical criteria and response to treatment in comparison to others with the same morphology classification and include a paragraph
Reviewer 2 Report
Comments and Suggestions for Authors
Materials and Methods:
Some descriptions are redundant, such as the protocols of therapy regimes that were already clearly described in Table 1. Also, some data belong to the results section (e.g. lines 194 - 202 in the Histopathological Diagnosis paragraph). This whole section should be revised.
Line 299: according to the National Cancer Institute, progression-free survival is defined as “the length of time during and after the treatment of a disease, that a patient lives with a disease but it does not get worse…”, and not from the diagnosis date as you wrote.
GELF criteria: Definition of low burden versus high burden disease need clarification. Based on reference 30, patients with spleen or liver enlargement, risk of loal compression, vital organ compression (data not provided for the different cFL subgroups) and leukemia (2,4% in ASD-LB in table 3) should be assigned to high burden disease.
Results:
Figure 2 is missing.
Table 4
The column “AS-HTB (III/IV) has N=160, though the N in the text, on which the calculations were made, is reported as 156.
It is unclear what the last two lines of this table refer to.
Conclusions:
The Authors stated that they identified predictors associated with decreased OS. Nevertheless, the Authors poorly commented these predictors in Conclusion.
References:
References 12 and 24 are the same
Reference 86 doesn't exist
Minor points:
In Advanced Stage - Low Tumor Burden (AS-LTB) paragraph please substitute “02”, “03” and “06” with “2”, “3” and “6”, respectively.
Lines 558, 574, 586 and 638: with “casuistic”, do you intend “case study”?
Please correct at lines 630 and 632 “early-stage” with ES, as reported throughout the text.
Reviewer 3 Report
Comments and Suggestions for Authors
1. The manuscript includes the conclusion information regarding Figure 2. Although Figure 2 is explained in the manuscript, it is visually incomplete and cannot be understood. Figure 2 should be rearranged and added to the manuscript.
2. Statistical information should be added below all Tables.
3. References in the text should be reviewed. There is missing and incorrect numbering in the reference order in the manuscript. For example, line 70, the same reference (reference number 1) is written twice, probably line 70 should be corrected to reference 2.
4. References should be written according to the journal format and should be numbered according to the same format in the manuscript.
5. All references should be checked.
6. There are spelling mistakes and word errors. It should definitely be reviewed and checked.
